# Development of Antibody–Drug Conjugates for Malignancies of the Uterine Corpus: A Review

**DOI:** 10.3390/cells14050333

**Published:** 2025-02-24

**Authors:** Taro Yamanaka, Tadaaki Nishikawa, Hiroshi Yoshida

**Affiliations:** 1Department of Medical Oncology, National Cancer Center Hospital, Tokyo 104-0045, Japan; tayaman2@ncc.go.jp; 2Department of Obstetrics and Gynecology, The Jikei University School of Medicine, Tokyo 105-8461, Japan; tnishika@jikei.ac.jp; 3Department of Diagnostic Pathology, National Cancer Center Hospital, Tokyo 104-0045, Japan

**Keywords:** endometrial carcinoma, uterine carcinosarcoma, uterine sarcoma, antibody–drug conjugate, HER2, TROP2, B7-H4, FRα, AXL, B7-H3

## Abstract

Despite recent advances in cancer treatment, the prognosis for uterine malignancies (carcinoma and sarcoma) requires further improvement. Antibody–drug conjugates (ADCs) have emerged as a novel class of anti-cancer therapeutic agents, and multiple ADCs have been approved for other types of cancer. In 2024, trastuzumab deruxtecan received approval from the US Food and Drug Administration for cancer types and became the first ADC approved for the treatment of uterine malignancies. Many ADCs are currently being investigated in uterine malignancies, and therefore, there is a need to gain a deeper understanding of ADCs. In this article, we aim to provide a comprehensive overview of the advancements in ADCs. The contents of this article include the structure and mechanism of action, an analysis of recent clinical trials, and expected future clinical questions. This article also focuses on uterine sarcoma, which is not often highlighted as a target for ADC treatment.

## 1. Introduction

Uterine malignancies include epithelial tumors such as endometrial carcinoma (EC) and uterine sarcoma such as leiomyosarcoma (LMS) and endometrial stromal sarcoma (ESS). Endometrial carcinoma (EC) is the most common gynecologic malignancy in postmenopausal women, with an increasing incidence and mortality [1]. The prognosis for endometrial cancer has improved in recent years, especially due to immune checkpoint inhibitors [2,3,4,5,6,7], but there are still significant unmet medical needs. Uterine sarcoma has an inferior prognosis among uterine malignancies, accounting for 4–9% of all uterine malignancies, and 40–50% of uterine sarcomas are LMS [8].

Antibody–drug conjugate (ADC) is a therapeutic drug with three main components: an antibody, a linker, and a payload [9]. It has become an especially important therapeutic modality in oncology in recent years. As of 15 August 2024, 378 ADCs have been reported, of which the FDA has approved 11; 217 are in clinical development, and 150 have been discontinued [10]. The number of ADCs has increased significantly in recent years. As of October 2023, there were 551 clinical trials in progress, of which 353 (64%) were related to ADCs that had already been approved, and 198 (36%) were related to new ADCs [11]. There are no drugs approved explicitly for uterine malignancies at the time of writing. However, trastuzumab deruxtecan (T-DXd) has been approved by the Food and Drug Administration (FDA) for HER2-positive solid tumors regardless of tumor type [12]. As described below, multiple clinical trials are currently in progress, and the importance of ADCs is expected to increase in treating uterine malignancies. ADCs offer a novel and targeted therapeutic strategy for uterine malignancies, addressing key limitations of current treatments. Despite advances in immune checkpoint inhibitors and chemotherapy, the prognosis for advanced and recurrent EC and uterine sarcomas remains poor, necessitating alternative approaches.

In this paper, we will discuss the mechanisms of the efficacy of antibody–drug conjugates against tumors and review the existing evidence for their use in the treatment of uterine malignancies. We will also explore ongoing clinical trials and preclinical trials, highlighting the progress of the development of ADCs as an emerging therapeutic strategy for patients with uterine malignancies. We aim to provide a narrative review on the development of ADCs for endometrial carcinoma and uterine sarcoma rather than conducting a comprehensive and exhaustive literature search. The details of individual clinical trials have also been referenced from ClinicalTrials.gov.

## 2. General Overview of ADC

### 2.1. Structure and Mechanism of Action

An ADC consists of three key components: a monoclonal antibody, a cytotoxic payload, and a linker that connects the two. The clinical efficacy of ADCs changes as each element changes. After administration, the ADC binds to the target antigen on the cancer cell surface, and then the ADC is internalized through clathrin-mediated endocytosis [13]. ADCs with cleavable linkers in the cancer cell release their payload into the cytoplasm, which induces cell death by causing DNA damage or inhibiting microtubule function. The intracellular payload may also cross the cell membrane and show an antitumor effect on surrounding cancer cells, called the “bystander effect” [14]. This phenomenon may explain the efficacy of ADCs on cancer cells in which the target antigens are not expressed. Since tumors are heterogeneous cell populations, the bystander effect is important. With these mechanisms, it is theoretically possible for ADCs to deliver cytotoxic agents to malignant cells effectively while minimizing side effects on non-target cells.

### 2.2. Antibody and Target Antigen

Monoclonal antibodies enable the ADCs to specifically bind to the target antigen on cancer cells. Humanized antibodies are often used as monoclonal antibodies for ADCs. Of the five immunoglobulin types (IgM, IgA, IgD, IgE, IgG), immunoglobulin G (IgG) is the most frequently used in ADCs. Furthermore, IgG1 is widely used because it usually supports antibody-dependent cellular cytotoxicity (ADCC) and complement-dependent cytotoxicity (CDC) [15]. The ideal target antigen for antibody–drug conjugates should be highly expressed on tumor cells, efficiently internalized via endocytosis upon antibody binding, and exhibit minimal expression in normal tissues [16].

### 2.3. Payload

Many cytotoxic agents used as payloads in ADCs are currently limited to microtubule inhibitors, topoisomerase I inhibitors, and DNA targeting agents [17]. This may raise concerns about potential cross-resistance among ADCs with similar payloads, impacting their sequential use. The drug-to-antibody ratio (DAR) represents the average number of payload molecules conjugated to the antibody and plays a crucial role in determining the efficacy of ADCs. At first glance, a higher DAR reflects a higher efficacy but can result in reduced efficacy due to increased hydrophobicity [18]. Therefore, identifying the optimal DAR is essential.

### 2.4. Linker

Linkers bind the antibody to the cytotoxic payload. Ideal linkers should maintain the ADC’s stability in plasma while ensuring payload release, specifically at the tumor site. This targeted release is essential for effectively delivering the cytotoxic payload to cancer cells, as premature release can result in systemic off-target toxicity [19]. Linkers can be classified into two types: cleavable and non-cleavable. Cleavable linkers release their payloads in response to specific conditions, such as the pH of cancer cells, reducing agents, and enzymes. Cleavable linkers may increase the efficiency of payload release, causing a bystander effect, but they also have a higher risk of increasing toxicity in normal tissues. On the other hand, non-cleavable linkers are more stable in plasma, so they only release the drug after the antibody has been completely degraded in the lysosome. ADCs with non-cleavable linkers are not expected to induce a bystander effect. Currently, most ADCs use cleavable linkers.

## 3. ADC for Endometrial Carcinoma

### 3.1. Overview of the Histological and Molecular Features of Endometrial Carcinoma (Figure 1)

EC has traditionally been classified into two categories—type I and type II—based on clinicopathological features, as proposed by Bokhman in 1983 [20]. Type I EC, which accounts for the majority of cases, includes low-grade (grade 1–2) endometrioid carcinomas with favorable prognoses. These tumors typically arise in estrogen-rich environments (e.g., obesity) and progress from atypical endometrial hyperplasia or endometrioid intraepithelial neoplasia, characterized by unopposed estrogen signaling and subclonal evolution [21,22,23]. Endometrioid carcinomas are graded according to the glandular-to-solid growth ratio following the FIGO system [24]. In contrast, type II ECs are hormone-independent, more aggressive, and associated with poor prognoses [20]. This category includes high-grade histologic subtypes such as serous carcinoma, clear cell carcinoma, carcinosarcoma, and undifferentiated/dedifferentiated carcinoma. Recently, mesonephric-like adenocarcinoma and gastric-type mucinous carcinoma have also been recognized as aggressive variants [25]. Serous carcinoma, the most common type II EC (up to 10% of cases), often arises from atrophic endometrium or polyps and is driven primarily by *TP53* mutations [26]. Despite limited local invasion, it frequently presents with occult metastases and accounts for approximately 40% of EC-related deaths [27,28]. Clear cell carcinoma (<5% of cases) is rare, resistant to chemotherapy, and associated with poor outcomes [29,30]. Carcinosarcomas (<5% of cases) are biphasic tumors with both epithelial and sarcomatous components, classified as homologous or heterologous, and exhibit aggressive clinical behavior [31,32,33]. Undifferentiated/dedifferentiated carcinomas are poorly differentiated, aggressive tumors characterized by monotonous cells, frequent SWI/SNF complex deficiencies (involving ARID1A, ARID1B, SMARCA4, and SMARCB1), and mismatch repair (MMR) deficiency [34,35,36,37]. Mesonephric-like adenocarcinoma shares histological features with mesonephric carcinoma of the uterine cervix, frequently harbors *KRAS* mutations and exhibits aggressive behavior [38,39,40]. While the type I/II classification provides foundational insights, it oversimplifies EC’s biological and clinical heterogeneity. Contemporary classification systems incorporate molecular profiling, offering a more nuanced understanding of tumor behavior and guiding personalized treatment strategies.

**Figure 1 cells-14-00333-f001:**
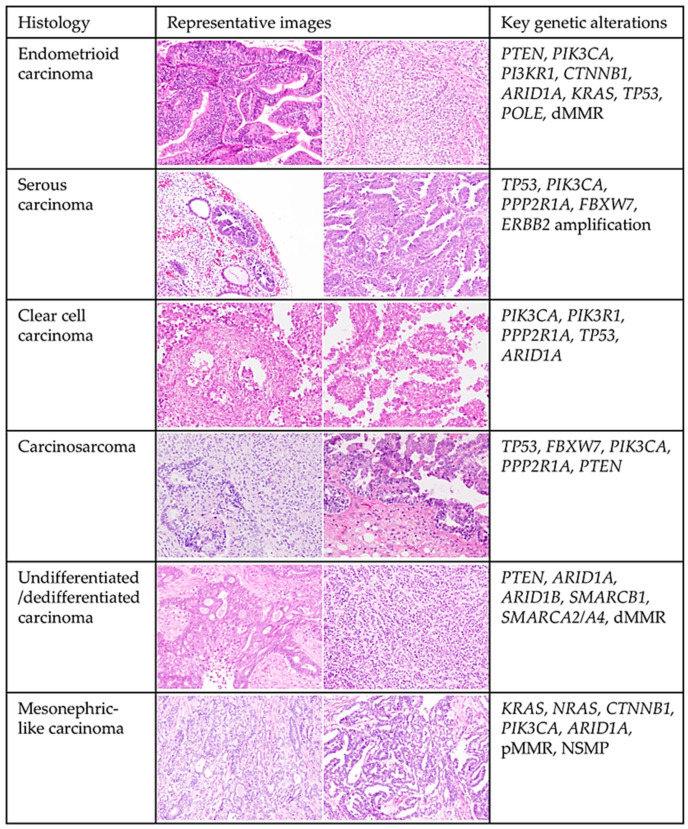
Histological and Molecular Features of Endometrial Carcinomas. Representative histopathological images illustrating the key histological subtypes of endometrial carcinoma, including endometrioid, serous, clear cell, carcinosarcoma, undifferentiated/dedifferentiated, and mesonephric-like carcinoma. Relevant molecular alterations associated with each subtype are also highlighted.

### 3.2. Molecular Characterization of Endometrial Cancer

The Cancer Genome Atlas (TCGA) has significantly advanced the molecular understanding of EC, classifying it into four distinct subtypes [41]: *POLE*-ultramutated, microsatellite instability-high (MSI-H), copy number-low, and copy number-high. The *POLE*-ultramutated subtype (~9% of ECs) is defined by pathogenic mutations in the exonuclease domain of the *POLE* gene, which plays a key role in DNA replication and repair [41,42,43]. These tumors exhibit an exceptionally high tumor mutational burden (TMB) and, despite often presenting as high-grade (grade 3) endometrioid carcinomas, are associated with excellent prognoses, likely due to robust immune infiltration [41,43]. The MSI-H subtype (~30% of ECs) is characterized by mismatch repair deficiency (MMRd), resulting in high MSI levels. MMRd is most commonly caused by *MLH1* promoter methylation (~80%), with less frequent contributions from Lynch syndrome (~10%) or somatic mutations (~20%) in MMR genes [41,44]. These tumors also show high TMB and immune infiltration, with intermediate prognoses. The copy number-high subtype includes most serous carcinomas and approximately 25% of high-grade endometrioid carcinomas. It is characterized by frequent *TP53* mutations and gene alterations such as *FBXW7*, *PPP2R1A*, *CCNE1*, and *PIK3CA*, contributing to aggressive behavior and poor prognosis [41]. The copy number-low subtype (~38% of ECs) primarily consists of low-grade endometrioid carcinomas with favorable outcomes. However, *CTNNB1* mutations, chromosome 1q amplifications, and estrogen receptor loss are associated with worse prognoses despite otherwise low-risk features [41,45]. Although TCGA’s molecular classification has provided valuable insights, its reliance on complex genomic analyses limits its feasibility in routine clinical practice due to cost and resource demands. To address this, the ProMisE (Proactive Molecular Risk Classifier for Endometrial Cancer) investigators developed a cost-effective, stepwise algorithm for classifying formalin-fixed paraffin-embedded (FFPE) EC samples based on MMR protein IHC, p53 IHC, and *POLE* mutation testing (Figure 2) [46,47]. This approach defines four molecular subtypes—*POLE*-mutant, MMR-deficient (MMRd), p53-abnormal, and NSMP—paralleling TCGA categories. The simplified methodology bridges the gap between advanced genomic research and routine clinical practice, supporting personalized treatment strategies based on molecular profiling. Importantly, the 2023 FIGO staging system for endometrial cancer has formally incorporated molecular classification into its framework [25]. This integration reflects the growing recognition of molecular features as critical determinants of prognosis and treatment response.

### 3.3. The Current State of Standard Systemic Therapies for Endometrial Carcinoma

For the first-line treatment of advanced or recurrent endometrial cancer, TC (Paclitaxel and Carboplatin) or AP (Doxorubicin plus Cisplatin) therapy has long been used as the standard of care. In the second-line treatment, immune checkpoint inhibitor monotherapy has demonstrated efficacy in patients with MMRd endometrial cancer, and the phase III KEYNOTE-775 study has established the combination of lenvatinib and pembrolizumab as the standard treatment for patients with both MMRd and MMR-proficient (MMRp) endometrial cancer [4,5]. Furthermore, in recent years, multiple phase III trials have reported the efficacy of immune checkpoint inhibitors combined with chemotherapy as a first-line treatment, leading to its establishment as a new standard of care [3,6,7,10]. The three trials, AtTEnd, RUBY, and DUO-E, all included patients with UCS [2,6,7], raising expectations for the efficacy of these therapies in UCS as well. Despite these advances in treatment, the prognosis remains suboptimal, highlighting the need for novel treatment strategies, including ADCs.

### 3.4. Promising Target for Endometrial Carcinoma

#### 3.4.1. HER2

1.Trastuzumab Deruxtecan

Trastuzumab deruxtecan (T-DXd) is an ADC composed of a humanized anti-HER2 monoclonal antibody conjugated to a topoisomerase I inhibitor payload via a tetrapeptide-based cleavable linker [48,49]. T-DXd has been approved for various types of previously treated advanced or recurrent cancers (HER2-positive breast and gastric cancer, HER2-low breast cancer, and HER2-mutated non-small cell lung cancer). Additionally, it received FDA approval for use in HER2-positive (immunostaining 3+) solid tumors based on the DESTINY-Pantumor02 study, making it the first ADC approved for use in endometrial cancer. The DESTINY-PanTumor 02 study was a phase II basket trial for patients with incurable solid tumors and HER2 immunostaining of 2+ or 3+. The study included 267 patients who had already received previous treatment, and 40 of these were patients with endometrial cancer. The overall response rate (ORR), which was the primary endpoint, was 57.5%, but when limited to 3+ cases, the response rate was extremely favorable at 84.6% (*n* = 13). Although the number of cases is small, and it is difficult to compare results by cancer type, the results for endometrial cancer were better than for other cancers (such as cervical cancer, ovarian cancer, and bladder cancer). The rate of discontinuation due to adverse events in the uterine cancer group was 7.5%, which was not much different from rates in other trials [50]. The STATICE trial (NCCH1615) was an investigator-initiated phase II clinical trial conducted in Japan to evaluate the efficacy and safety of T-DXd in patients with unresectable uterine carcinosarcoma who had a history of previous systemic therapy. The main analysis included patients with HER2 protein expression levels of 2+ or 3+ (HER2-high), while patients with HER2 expression of 1+ (HER2-low) were included in the exploratory analysis. A total of 22 patients were enrolled in the HER2-high group and 10 patients in the HER2-low group. The ORR was 54.5% for the HER2-high group and 70% for the HER2-low group, demonstrating highly favorable outcomes [51].

2.Trastuzumab Duocarmazine

Trastuzumab duocarmazine is a HER2-directed ADC composed of a monoclonal IgG1 antibody trastuzumab linked to a DNA alkylating agent payload (duocarmycin) via a cleavable linker. In the dose expansion cohort of the phase I trial (NCT02277717), patients with HER2 IHC 1+ or greater breast cancer, gastric cancer, urothelial cancer, or endometrial cancer were included. Of the 13 patients with endometrial cancer, 5 showed an objective response (39%), with a median PFS of 4.3 months. The most common treatment-related adverse events (TRAE) were fatigue, conjunctivitis, and dry eye [52]. A phase II NCT04205630 trial to evaluate the efficacy of trastuzumab duocarmazine in patients with HER2-expressing, advanced endometrial cancer has completed the enrollment, and the results are awaited.

3.BNT323/DB-1303

DB-1303 is a HER2-directed ADC composed of a humanized anti-HER2 IgG1 monoclonal antibody linked to a topoisomerase I inhibitor payload (P1003) via a maleimide tetrapeptide-based cleavable linker [53]. In the dose escalation and dose expansion cohorts of the Phase 1 study (NCT05150691), objective partial tumor responses were observed in 10 (58.8%) of the 17 patients with endometrial cancer who received BNT323/DB-1303 and were evaluable for response. The most common TRAE were nausea, fatigue, and vomiting [54]. A phase III ENGOT-EN25/BNT323-01/NSGO trial (NCT06340568) to evaluate the efficacy of DB-1303 versus the investigator’s choice of chemotherapy (ICC) will start.

#### 3.4.2. TROP2

Trophoblast cell surface antigen 2 (Trop-2), an epithelial antigen overexpressed in about 57% of endometrioid endometrial cancer, is associated with poor prognosis and higher tumor grades [55], representing a potential treatment target.

1.Sacituzumab Govitecan

Sacituzumab govitecan (SG) is an ADC with Trop-2 antibodies coupled to a cytotoxic SN-38 payload via a proprietary hydrolyzable linker [56,57]. The FDA and EMA have already approved SG for the treatment of patients with pre-treated, advanced triple-negative breast cancer (TNBC) and hormone receptor-positive and HER2-negative breast cancer based on the phase III ASCENT and TROPiCS-02 trial. In the two phase III trials, SG demonstrated statistically significant and clinically meaningful benefits over ICC [58,59]. The most common grade ≥3 adverse events associated with SG were neutropenia, leukopenia, diarrhea, anemia, and febrile neutropenia [58]. The TROPiCS-03 trial was a phase II basket study in patients with metastatic solid tumors, including endometrial cancer. Among 41 patients with endometrial cancer, a confirmed objective response was observed in 22% (n = 9; 95% CI, 11 to 38) of patients, based on investigator assessment [60]. The Phase III ASCENT-GYN-01/GOG-3104/ENGOT-en26 trial is currently ongoing to evaluate the efficacy of SG versus ICC in patients with endometrial cancer who have previously received treatment with platinum-based chemotherapy and immunotherapy (NCT06486441).

2.Datopotamab Deruxtecan

Datopotamab deruxtecan (Dato-DXd) is a TROP2-directed ADC composed of a humanized anti-TROP2 IgG1 monoclonal antibody linked to a highly potent topoisomerase I inhibitor payload (an exatecan derivative, DXd), via a tetrapeptide-based cleavable linker [61]. In the phase III TROPION-Breast01 study for patients with hormone receptor-positive/human epidermal growth factor receptor 2–negative (HR+/HER2−) breast cancer, Dato-DXd significantly reduced the risk of progression or death versus ICC (PFS by BICR hazard ratio [HR], 0.63 [95% CI, 0.52 to 0.76]; *p* < 0.0001). The most common TRAE were nausea and stomatitis. The grade ≥3 TRAEs with Dato-DXd rate was less than half that with ICC [62]. On 17 January 2025, based on the TROPION-Breast01 trial, the FDA approved Dato-DXd for adult patients with unresectable or metastatic, hormone receptor (HR)-positive, human epidermal growth factor receptor 2 (HER2)-negative (IHC 0, IHC1+ or IHC2+/ISH−) breast cancer who have received prior endocrine-based therapy and chemotherapy for unresectable or metastatic disease [63]. Dato-DXd is currently being evaluated in the phase II TROPION-Pantumor03 study (NCT05489211) in patients with advanced solid tumors, including endometrial cancer.

3.Sacituzumab Tirumotecan

Sacituzumab tirumotecan is a TROP2-targeting ADC consisting of a humanized IgG1/κ monoclonal antibody conjugated to a belotecan-derivative topoisomerase I inhibitor tirumotecan (KL610023) via a cleavable, pH-sensitive linker [64]. Sacituzumab tirumotecan was approved by China’s National Medical Products Administration (NMPA) on 27 November 2024 for adult patients with unresectable locally advanced or metastatic TNBC, based on the results of the phase III OptiBreast01 trial. In this trial, sacituzumab tirumotecan demonstrated statistically significant and clinically meaningful benefits compared to ICC. The most common grade ≥3 treatment-related adverse events included neutropenia, anemia, and white blood cell count [65]. Sacituzumab tirumotecan is currently being evaluated in the phase III MK-2870–005/ENGOT-en23/GOG-3095 trial comparing sacituzumab tirumotecan monotherapy with ICC in patients with endometrial cancer who have previously undergone treatment with platinum-based chemotherapy and immunotherapy (NCT06132958).

#### 3.4.3. FRα

Folate receptor-alpha (FRα) is a cell-surface glycoprotein that binds and transports physiological levels of folate with high affinity. FRα is expressed in various epithelial tumors [66], including EC [67], and its expression has been linked to poor prognosis in EC [68,69].

1.Mirvetuximab Soravtansine (MIRV)

Mirvetuximab soravtansine is an FRα-targeting ADC composed of a humanized IgG1/κ monoclonal antibody linked to a tubulin-disrupting maytansinoid DM4 via a cleavable linker [70]. The FDA currently approves MIRV for the treatment of adult patients with FRα-positive, platinum-resistant epithelial ovarian, fallopian tube, or primary peritoneal cancer who have received one to three prior systemic treatment regimens [71]. In the dose escalation cohort of the phase I trial (NCT01609556), clinical benefit was observed in two out of 11 patients with EC [72]. MIRV is currently being investigated as a monotherapy in a phase II trial for patients with FRα-positive persistent or recurrent endometrial cancer (NCT03832361). Recently, the results of a phase II trial examining the efficacy of MIRV combined with pembrolizumab for patients with advanced or recurrent microsatellite stable/MMRp, FRα-positive serous EC were reported. Among the 16 patients treated, 6 (37.5%) achieved an objective response. The most common TRAEs included AST elevation (50%), blurred vision (44%), fatigue (44%), and diarrhea (43%) [68].

2.Farletuzumab Ecteribulin (MORAb-202)

Farletuzumab ecteribulin is an FRα-targeting ADC consisting of a humanized monoclonal antibody conjugated to a microtubule inhibitor (eribulin) via a cathepsin-B cleavable linker [73]. In a phase I study, one of the three patients with endometrial cancer showed a partial response, and the remaining two experienced stable disease. The most common TRAEs included leukopenia, neutropenia, ALT increased, anemia, and AST increase. Farletuzumab ecteribulin is now being investigated in phase I/II trials for adult patients with solid tumors, including EC (NCT04300556).

3.Luveltamab Tazevibulin

Luveltamab tazevibulin is an FRα-targeting ADC consisting of a humanized monoclonal antibody conjugated to a microtubule inhibitor (SC209) via a cathepsin-cleavable linker [74]. In the dose expansion cohort of the phase I trial, 6 out of 16 evaluable patients with EC (37.5%) showed a response. The most common ≥ Grade 3 AEs included neutropenia, anemia, and arthralgia [74]. Luveltamab tazevibulin is currently under investigation in a phase I/II trial for Chinese adult patients with solid tumors, including EC (NCT06238687).

4.Rinatabart Sesutecan

Rinatabart sesutecan is an FRα-targeting ADC consisting of a humanized monoclonal antibody conjugated to a topoisomerase I inhibitor (exatecan) via a cleavable hydrophilic linker [75]. In the ongoing phase I/II PRO1184-001 trial, the most common grade ≥ 3 TRAEs occurring in ≥10% of patients were neutropenia, anemia, leukopenia, and thrombocytopenia. Promising antitumor activity was observed across a broad spectrum of FRα expression levels [75].

#### 3.4.4. B7-H4

B7-H4 is a transmembrane glycoprotein belonging to the B7 superfamily that negatively regulates T cell function [76]. It is overexpressed in many types of cancers, with EC being one of the solid tumors exhibiting the highest expression levels of B7-H4 (94%) [77]. The clinical significance of B7-H4 is underscored by its high expression in numerous tumor tissues and its association with adverse clinical and pathological features, such as increased tumor aggressiveness [78].

1.SGN-B7H4V

SGN-B7H4V is a B7-H4-targeting ADC composed of a humanized IgG1 monoclonal antibody linked to a microtubule-disrupting agent, monomethyl auristatin E (MMAE), via a protease-cleavable maleimidocaproyl valine-citrulline linker [79]. Preliminary results from the dose-escalation phase of the ongoing phase I SGNB7H4V-001 trial (NCT05194072) identified fatigue, peripheral sensory neuropathy, and neutropenia as the most common treatment-emergent adverse events (TEAEs). One out of 16 patients with endometrial cancer achieved a confirmed complete response. The authors concluded that SGN-B7H4V demonstrated a manageable safety profile in patients with advanced solid tumors. A dose expansion cohort targeting specific tumor types was planned [80].

2.AZD8205 (Puxitatug Samrotecan)

AZD8205 is a B7-H4-directed ADC consisting of a monoclonal antibody linked to a topoisomerase I inhibitor (AZ14170133) via a cleavable linker [77]. In the initial results from the dose escalation part of the Phase I/IIa BLUESTAR trial (NCT05123482), the most common TEAEs were nausea, neutropenia, and anemia. Among 43 patients treated with doses ≥1.6 mg/kg, nine patients with ovarian, breast, or endometrial cancer achieved a confirmed partial response (20.9%). Phase II expansion cohorts are actively underway in ovarian, breast, endometrial, and biliary tract cancer [81].

3.XMT-1660 (Emiltatug Ledadotin)

XMT-1660 is a B7-H4-targeting ADC consisting of a humanized IgG1 monoclonal antibody linked to a microtubule inhibitor (auristatin F) via a cleavable linker [82]. It is currently being investigated in a phase 1 study (NCT05377996) for patients with advanced solid tumors, including endometrial cancer. The FDA has given XMT-1660 a new fast-track designation for the treatment of patients with advanced or metastatic HER2-low or HER2-negative breast cancer who have previously received a topoisomerase–1–directed ADC.

4.HS-20089

HS-20089 is a B7-H4-targeting ADC consisting of a humanized IgG1 monoclonal antibody linked to a topoisomerase I inhibitor via a protease-cleavable linker [83]. Preliminary results from the ongoing phase I trial (NCT05263479) showed that 44 patients, including one with endometrial cancer, received HS-20089, with a promising response rate of 24.2%. The most common TEAEs were leukopenia, neutropenia, nausea, anemia, thrombocytopenia, vomiting, fatigue, increased alanine aminotransferase, anorexia, increased aspartate aminotransferase and hyponatremia [83]. A Phase II trial (NCT06014190) is currently in progress only in China.

#### 3.4.5. Other Targets

Other ADCs are summarized in Table 1.

## 4. ADC for Uterine Sarcoma

### 4.1. Overview of the Histological and Molecular Features of Uterine Sarcoma (Figure 3)

Uterine sarcomas are rare malignant tumors, accounting for approximately 3–7% of all uterine malignancies [84,85]; their pathological diagnosis is guided by the WHO classification of tumors, which provides standardized criteria for histopathological evaluation [86]. Uterine sarcomas can present as pure mesenchymal or biphasic tumors with mesenchymal and epithelial components, as seen in adenosarcomas.

Among pure mesenchymal sarcomas, leiomyosarcoma (LMS) is the most common, representing approximately 60% of all uterine sarcomas [85]. LMS can be classified into conventional/spindle cell, epithelioid, and myxoid subtypes, each with distinct morphological features and grading criteria [86]. In addition to LMS, other types of uterine sarcomas include endometrial stromal sarcomas (ESS), which are subdivided into low-grade (LGESS) and high-grade (HGESS), undifferentiated uterine sarcoma (UUS), malignant perivascular epithelioid cell tumor (PEComa), and inflammatory myofibroblastic tumor (IMT) [86]. LGESS is the second most common uterine sarcoma after LMS and is reportedly characterized by specific gene rearrangements such as *JAZF1::SUZ12*, *JAZF1::PHF1*, and *EPC1::PHF1* [87]. In contrast, HGESS is associated with distinct genetic alterations, including *YWHAE::NUTM2A/B* and *BCOR* gene rearrangements or internal tandem duplications (ITDs), contributing to its aggressive clinical behavior [86]. Recent studies have redefined the nature of adenosarcomas, revealing that genetic alterations are typically absent in the epithelial component, confirming that these tumors are primarily mesenchymal malignancies with benign epithelial elements [88]. The pathological diagnosis of uterine sarcomas is often challenging due to overlapping histological features [89,90]. Additionally, emerging research has identified extremely rare uterine sarcoma entities associated with specific molecular alterations, such as *NTRK* gene fusions [91,92] and SMARCA4 deficiencies [93]. These discoveries have proposed new sarcoma classifications, reflecting the evolving understanding of uterine sarcoma pathogenesis. They also underscore the critical role of molecular diagnostics in enhancing diagnostic accuracy and guiding targeted therapies.

**Figure 3 cells-14-00333-f003:**
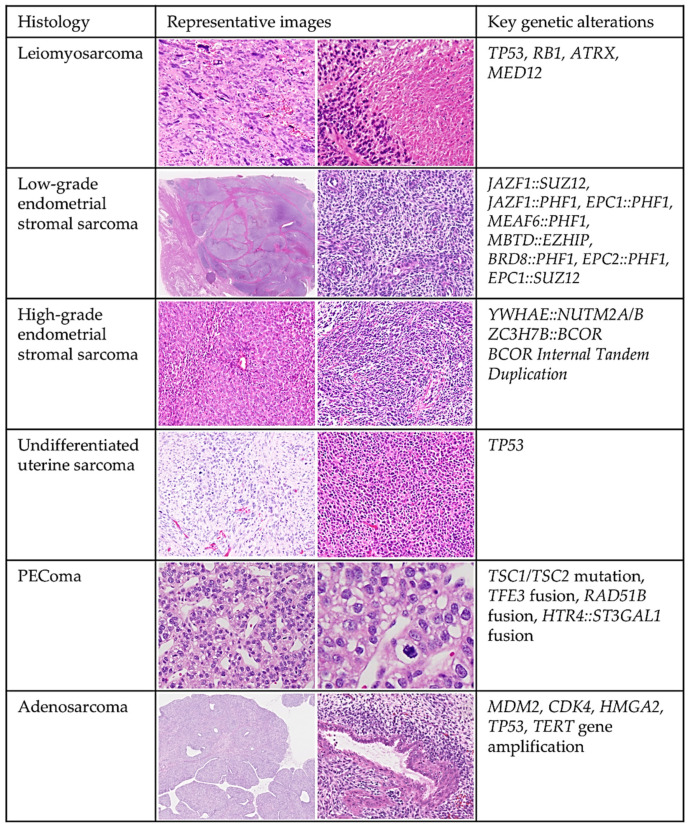
Histological and Molecular Features of Uterine Sarcomas. Representative histopathological images illustrating the key histological subtypes of uterine sarcoma, including leiomyosarcoma, low-grade endometrial stromal sarcoma, high-grade endometrial stromal sarcoma, undifferentiated uterine sarcoma, PEComa, and adenosarcoma. Relevant molecular alterations associated with each subtype are also highlighted.

### 4.2. The Current State of Standard Systemic Therapies for Uterine Sarcoma

The GeDDiS study, a phase III trial comparing gemcitabine plus docetaxel (GD) with doxorubicin in patients with advanced/recurrent soft tissue sarcoma, was unsuccessful in demonstrating the superiority of GD to doxorubicin. About 25% of the cases in this study were leiomyosarcoma [94]. Doxorubicin was the first-line treatment for advanced and recurrent disease for a long time, but the results of the LMS-04 trial, reported in 2024, changed the standard treatment. In this study, doxorubicin monotherapy and doxorubicin in combination with trabectedin followed by trabectedin maintenance therapy were compared in patients with advanced or recurrent leiomyosarcoma (about 45% of whom had uterine leiomyosarcoma). A significant extension of PFS, the primary endpoint, was observed (HR 0.41, 95% CI 0.29–0.58) in the doxorubicin in combination with trabectedin group [95]. Additionally, the phase II NCI Protocol 10250 study, which examined the efficacy and safety of temozolomide combined with olaparib, met its primary endpoint of response rate within 6 months, and the results met expectations [96]. On the other hand, scarce clinical trials regarding ADCs for sarcomas have been conducted.

### 4.3. Promising Targets for Uterine Sarcoma

Promising ADCs for uterine sarcoma are listed in Table 2.

#### 4.3.1. AXL

AXL is a member of the receptor tyrosine kinases family [97]. It is highly expressed in several sarcoma subtypes, where it has been linked to tumor resistance to chemotherapy [98]. AXL is also commonly expressed in both uterine and nonuterine LMS, with its expression being associated with poor prognosis [99].

1.Mecbotamab Vedotin (BA3011)

Mecbotamab vedotin is an AXL-targeting ADC consisting of a humanized IgG1 monoclonal antibody linked to a microtubule-disrupting agent monomethyl auristatin E (MMAE) via a cleavable linker [98]. In the phase 2 trial (NCT03425279) for adult and adolescent patients with AXL-expressing advanced refractory sarcoma, 87 patients received mecbotamab vedotin monotherapy, and 26 patients received mecbotamab vedotin plus nivolumab. The most common TEAEs of special interest among monotherapy patients included peripheral neuropathy, neutropenia, and abnormal liver function tests. The disease control rate was 41.1%, while the response rate was 4.5% across all patients.

2.Enapotamab Vedotin

Enapotamab vedotin is an AXL-targeting ADC composed of a humanized IgG1 monoclonal antibody linked to a microtubule-disrupting agent monomethyl auristatin E (MMAE) via a protease cleavable valine-citrulline linker [100]. Enapotamab vedotin showed preliminary antitumor activity in the phase I/II trial for relapsed/refractory solid cancers, including endometrial cancer [101].

#### 4.3.2. B7-H3

B7-H3 is a member of the B7 ligand family and inhibits tumor antigen-specific immune responses, contributing to a protumorigenic effect in malignant tissues. Consequently, B7-H3 expression in tumors is associated with poor prognosis [102]. In the recent report, B7-H3 is highly expressed in a variety of soft tissue sarcomas, including LMS (97%), liposarcoma (100%), and undifferentiated sarcoma (96%) [103].

Ifinatamab Deruxtecan (DS7300a)

Ifinatamab deruxtecan is a B7-H3-directed ADC composed of a monoclonal antibody linked to a topoisomerase I inhibitor (DXd) via a stable cleavable linker [104]. Ifinatamab deruxtecan is evaluated in the phase 1b/2 Ideate-Pantumor02 trial (NCT06330064) for patients with recurrent or metastatic solid tumors, including EC.

#### 4.3.3. GD2

GD2 is a cell membrane component found in a limited number of normal tissues but is expressed in certain tumor types, including neuroblastoma, osteosarcoma, glioma, and soft tissue sarcoma. In a study that examined 11 uterine leiomyosarcoma specimens using immunohistochemical staining, GD2 expression was observed in all tumors [105].

M3554

M3554 is a GD2-directed ADC composed of a monoclonal antibody linked to a topoisomerase I inhibitor (exatecan) via a cleavable beta-glucuronide linker [106]. M3554 is currently evaluated in a phase I trial (NCT06641908) in patients with soft tissue sarcoma and glioblastoma, IDH-wildtype.

#### 4.3.4. Leucine-Rich Repeat Containing 15

Leucine-rich repeat containing 15 (LRRC15) is a membrane protein found on the cell surface of stromal fibroblasts in various solid tumors, while its expression is low in most normal tissues [107].

ABBV-085

M3554 is a LRRC15-directed ADC consisting of a humanized IgG1 monoclonal antibody linked to a microtubule-disrupting agent monomethyl auristatin E (MMAE) via a protease-cleavable valine-citrulline linker. ABBV-085 demonstrated preliminary antitumor activity in patients with osteosarcoma and undifferentiated pleomorphic sarcoma [107].

## 5. Future Directions

### 5.1. Mechanisms of Resistance to ADCs

As noted earlier, the selection of cases for ADC administration is often based on the expression levels of target proteins. However, the factors predicting the efficacy of ADCs are still largely unknown. Given the limited research on the resistance mechanisms in uterine cancer, this review will also incorporate findings from other cancer types.

The efficacy and safety of SG for patients with endometrial cancer were examined in the TROPiCS-03 study [60]. TROP2 expression (>0%) was detected in 92% of patients, and its relationship with antitumor effects was examined. Subgroup analysis of PFS according to Trop-2 expression in tumors showed similar efficacy, with a median PFS of 4.2 months in the subgroup with a median or higher Trop-2 H-score (≥115) and 5.0 months in the subgroup with a median or lower Trop-2 H-score (<115) (hazard ratio, 0.9; 95% CI, 0.4 to 2.0; *p* = 0.8). The expression of Trop-2 was found to have a limited correlation with the efficacy of SG [60].

In the phase III ASCENT trial evaluating the efficacy and safety of SG in patients with previously treated TNBC, patients with low TRPO2 expression (H-score: 0–130) had a lower ORR than patients with moderate to high expression (H-score ≥130), but the efficacy of SG was found to be superior to that of the standard treatment group in all levels of TROP-2 expression. While TROP2 expression may be involved in the resistance mechanism to SG, it is thought that SG showed efficacy even in tumors with low TROP2 expression due to the bystander effect, high binding affinity, and high drug-antibody ratio [108].

Enfortumab vedotin (EV), an ADC targeting nectin4, has received FDA approval for treating previously treated urothelial carcinoma based on the results of the EV-301 study. In addition, the EV-302 study, for previously untreated advanced or recurrent urothelial carcinoma, showed the superiority of pembrolizumab plus EV over platinum combination therapy, which had long been the standard treatment for first-line treatment and has been established as a new standard treatment. These regimens can be administered regardless of nectin-4 expression on tumor specimens. On the other hand, in a retrospective study that examined the relationship between the level of *NECTIN4* amplification in tumor tissue and prognosis in patients receiving EV monotherapy, nonamplified cases were found to have significantly shorter PFS and OS than amplified cases (*p* < 0.001). *NECTIN4* amplification has been reported to predict the genomic effects of EVs in metastatic urothelial carcinoma [109].

The mechanism may differ depending on the type of drug or cancer, and further investigation is needed into the mechanism of resistance in malignant uterine malignancies.

### 5.2. Sequencing Strategies

As clinical trials are conducted on many different ADCs, clinical questions may arise regarding the optimal strategies for their use once multiple ADCs become available. One key consideration is determining the most effective sequence of administration when several ADCs are options for treatment. This section uses breast cancer as an example of a solid tumor currently treatable with multiple ADCs.

Breast cancer is classified into several subtypes according to hormone receptor and HER2 status, and each subtype has a different treatment strategy. SG and T-DXd are available for the treatment of previously treated metastatic or recurrent triple-negative and HER2-low breast cancer. In addition, Dato-DXd, SG, and T-DXd are available for metastatic or recurrent hormone receptor-positive and HER2-low breast cancer. Key issues under discussion include (1) whether ADCs with the same payload remain effective after another ADC with a similar payload becomes ineffective, (2) whether ADCs targeting the same antigen but using different payloads remain effective after resistance to one occurs, and (3) what the optimal sequence of administration should be. Retrospective studies have suggested that progression-free survival with a second ADC is often shorter than with the first ADC [110,111].

Several ongoing clinical trials are addressing these questions. The phase II TRADE-DXd study (NCT06533826) is evaluating the sequential administration of T-DXd and Dato-DXd, both of which use a topoisomerase I inhibitor as their payload (T-DXd followed by Dato-DXd or vice versa). Additionally, the SERIES trial (NCT06263543) is being conducted to investigate the efficacy of SG following T-DXd administration in patients with hormone receptor-positive, HER2-low breast cancer. The findings from these trials will be important to consider.

Similar clinical questions are expected to arise in uterine malignancies, making it essential to consider planning and conducting multiple clinical trials to address these potential scenarios.

## 6. Conclusions

This review provides an overview of the latest information on ADCs currently in clinical use and those under development for uterine malignancies. While the development of ADCs for EC is progressing rapidly, the development of ADCs for sarcomas appears to be limited. Furthermore, as the efficacy of multiple ADCs is demonstrated, new clinical questions regarding optimal treatment strategies are likely to emerge. Continued efforts in treatment development are urgently needed to ensure ADCs are delivered promptly and appropriately to patients with uterine malignancies.

## Figures and Tables

**Figure 2 cells-14-00333-f002:**
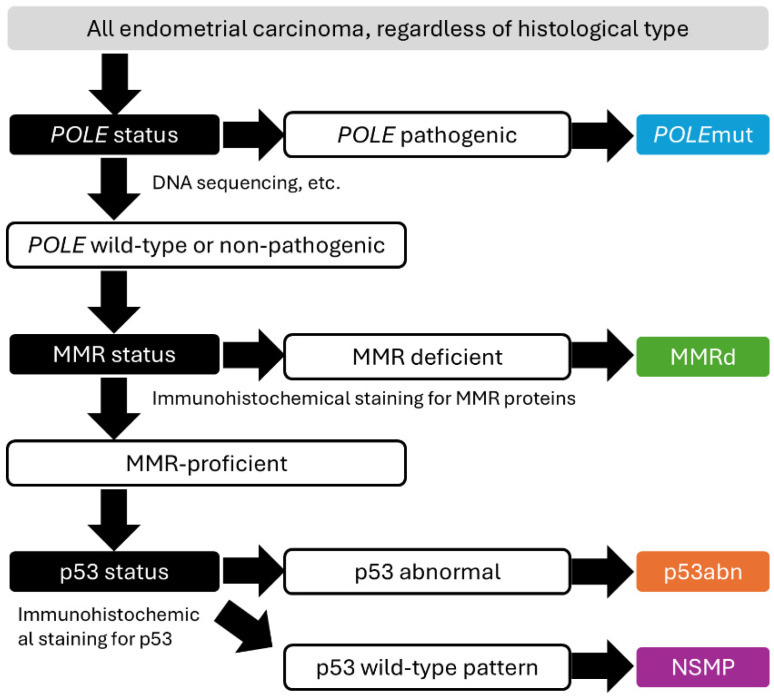
The TCGA-surrogate approach of molecular classification of endometrial carcinoma. Schematic representation of the TCGA-surrogate approach for molecular classification of endometrial carcinomas. The four molecular subtypes—POLE-mutant (POLE), mismatch repair-deficient (MMRd), p53-abnormal (p53abn), and non-specific molecular phenotype (NSMP)—are depicted.

**Table 1 cells-14-00333-t001:** Antibody–drug conjugates for endometrial carcinomas.

Antibody Target	Agent	Payload	Type of Payload	DAR	Linker	Phase	Clinical Trials
HER2	Trastuzumab deruxtecan	Deruxtecan	Topoisomerase I inhibitor	8	Cleavable	II	NCT04482309
HER2	Trastuzumab duocarmazine	Duocarmazine	DNA targeting agent	2.8	Cleavable	II	NCT04205630
HER2	BNT323/DB-1303	P1003	Topoisomerase I inhibitor	8	Cleavable	I/IIA	NCT05150691
						III	ENGOT en25/GOG 3105/BNT323 01
HER2	Disitamab vedotin	MMAE	Anti microtubule agent	4	Cleavable	I	NCT02881190
HER2	IBI354	NT3	Topoisomerase I inhibitor	8	Cleavable	I	NCT05636215
TROP2	Sacituzumab govitecan	SN-38	Topoisomerase I inhibitor	7	Cleavable	III	ASCENT-GYN-01/GOG-3104/ENGOT-en26
TROP2	Sacituzumab tirumotican	Tirumotican	Topoisomerase I inhibitor	7.4	Cleavable	III	MK-2870–005/ENGOT-en23/GOG-3095
TROP2	Datopotamab deruxtecan	Deruxtecan	Topoisomerase I inhibitor	4	Cleavable	II	NCT05489211
FRα	Mirvetuximab soravtansine	DM4	Anti-tubulin maytansinoid agent	3.5	Cleavable	II	NCT03832361
FRα	Farletuzumab ecteribulin (MORAb202)	Eribulin mesylate	Anti microtubule agent	4	Cleavable	I/II	NCT04300556
FRα	Luveltamab tazevibulin (STRO-002)	Hemiasterlin	Anti microtubule agent	4	Cleavable	I/II	NCT03748186, NCT06238687
FRα	Rinatabart sesutecan (Rina S, PRO1184)	Exatecan	Topoisomerase I inhibitor	8	Cleavable	I/II	NCT05579366
FRα	IMGN151	DM21	Anti-tubulin maytansinoid agent	3.5	Cleavable	I	NCT05527184
FRα	LY4170156	Exatecan	Topoisomerase I inhibitor	8	Cleavable	I	NCT06400472
B7-H4	AZD8205	AZ14170133	Topoisomerase I inhibitor	8	Cleavable	I/IIA	NCT05123482
B7-H4	HS-20089	-	Topoisomerase I inhibitor	6	Cleavable	I	NCT05263479
B7-H4	SGN-B7H4V	MMAE	Anti microtubule agent	4	Cleavable	I	NCT05194072
B7-H4	XMT-1660	Auristatin F	Anti microtubule agent	6	Cleavable	I	NCT05377996
TF	Tisotumab vedotin	MMAE	Anti microtubule agent	4	Cleavable	II	
Claudin6	TORL-1-23	MMAE	Anti microtubule agent	~4	Cleavable	I	NCT05103683
Nectin4	LY4101174	Exatecan	Topoisomerase I inhibitor	8	Cleavable	I	NCT06400472

**Table 2 cells-14-00333-t002:** Antibody–drug conjugates for uterine sarcomas.

Antibody Target	Agent	Payload	Type of Payload	DAR	Linker	Phase	Clinical Trials
AXL	Mecbotamab vedotin (BA3011)	MMAE	Anti microtubule agent	1.8	Cleavable	I/II	NCT03425279
AXL	Enapotamab vedotin	MMAE	Anti microtubule agent	4	Cleavable	I/II	NCT02988817
B7-H3	Ifinatamab deruxtecan (DS7300a)	Deruxtecan	Topoisomerase I inhibitor	4	Cleavable	IB/II	NCT06330064
GD2	M3554	Exatecan	Topoisomerase I inhibitor	2	Cleavable	I	NCT06641908
LRRC15	ABBV-085	MMAE	Anti microtubule agent	2	Cleavable	I	NCT02565758

## Data Availability

Not applicable.

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
