# Peer review of "Development of Antibody–Drug Conjugates for Malignancies of the Uterine Corpus: A Review"

_cells, 2025, doi:10.3390/cells14050333_

Round 1

Reviewer 1 Report

Comments and Suggestions for Authors

I my oppinion, there are a large numbers of ADCs in detriment of their description and application. I would like to see more about the mainly ADCs. 

Author Response

Comment #1. I my opinion, there are a large numbers of ADCs in detriment of their description and application. I would like to see more about the mainly ADCs. 

Response #1. We sincerely appreciate your constructive comments. This review aims to provide a broad and comprehensive overview of various antibody-drug conjugates (ADCs) rather than to delve into the details of individual ADCs. However, we have included specific details on certain drugs that have already been approved or are in advanced stages of development. We greatly appreciate your time and effort in reviewing our manuscript and providing valuable comments.

Reviewer 2 Report

Comments and Suggestions for Authors

In the present review, authors summaryzed the latest information on ADCs currently being used in clinical  practice and ADCs under development for uterine malignancies. The development of ADCs for EC is progressing rapidly, although the development of ADCs for sarcomas appears to be limited. 

The manuscript is well written and data are clearly presented. 

In my opinion, it is suitable for pubblication. 

I have only a few suggestions:

1) A mor detailed overview on the histological and molecular features (TCGA) of endometrial carcinoma could be provided.

2) Amore detailed overview of endometrial sarcomas could be proivided: - endometrial stromal tumors (low-grade and high-grade endometrial stromal sarcomas); - Leiomyosarcoma (spindle cell, epithelioid, myxoid); - Uterine PEComas; Undifferentiated sarcoma

Author Response

In the present review, authors summaryzed the latest information on ADCs currently being used in clinical  practice and ADCs under development for uterine malignancies. The development of ADCs for EC is progressing rapidly, although the development of ADCs for sarcomas appears to be limited. The manuscript is well written and data are clearly presented. In my opinion, it is suitable for publication. 

Comment #1. I have only a few suggestions: A more detailed overview on the histological and molecular features (TCGA) of endometrial carcinoma could be provided. A more detailed overview of endometrial sarcomas could be proivided: - endometrial stromal tumors (low-grade and high-grade endometrial stromal sarcomas); - Leiomyosarcoma (spindle cell, epithelioid, myxoid); - Uterine PEComas; Undifferentiated sarcoma

Response #1. We sincerely appreciate your thoughtful review and constructive feedback on our manuscript. Your comments have been invaluable in improving the clarity and comprehensiveness of our work. We entirely agree with your suggestions. Accordingly, we have expanded the section on endometrial carcinoma to provide a more detailed overview of its histological and molecular characteristics, including a discussion of the TCGA classification. Furthermore, we have also added a more detailed description of endometrial sarcomas, covering endometrial stromal tumors (low-grade and high-grade endometrial stromal sarcomas), leiomyosarcoma (spindle cell, epithelioid, myxoid), uterine PEComas, and undifferentiated sarcoma. Additionally, we have updated the corresponding figures to reflect these additions. All the added text in the revised manuscript is highlighted in yellow. Again, we greatly appreciate your time and effort in reviewing our manuscript and providing valuable suggestions.

Reviewer 3 Report

Comments and Suggestions for Authors

Dear Authors,

Very interesting review paper regarding the use of Antibody Drug Conjugates. The study is well organized and comprehensively described, provides significant contribution in this field. 

One suggestion: please add few-sentence description of Your methodology at the end of Introduction sections. Please discuss whether You used FDA data to present the clinical trials with ADC or additionally You performed Pubmed database search? If FDA data, did You analyzed same corresponding European/Japanese database? Did You analyzed ClinicalTrials.gov website search to check the current status of each trial? The readers of Your paper need to be informed about Your paper preparation.

Author Response

Very interesting review paper regarding the use of Antibody Drug Conjugates. The study is well organized and comprehensively described, provides significant contribution in this field. 

Comment #11 One suggestion: please add few-sentence description of Your methodology at the end of Introduction sections. Please discuss whether You used FDA data to present the clinical trials with ADC or additionally You performed Pubmed database search? If FDA data, did You analyzed same corresponding European/Japanese database? Did You analyzed ClinicalTrials.gov website search to check the current status of each trial? The readers of Your paper need to be informed about Your paper preparation. 

Response #1. Thank you very much for your thoughtful review and for recognizing the significance of our work in this field. We truly appreciate your valuable feedback. Regarding the methodology, as this is a narrative review article, a detailed methodological description is not necessary, which aligns with the approach taken by many other narrative reviews. However, to enhance clarity for readers, we have briefly mentioned our methodology at the end of the Introduction section as follows: “We aim to provide a narrative review on the development of ADCs for endometrial carcinoma and uterine sarcoma, rather than conducting a comprehensive and exhaustive literature search. The details of individual clinical trials have also been referenced from ClinicalTrials.gov.” Again, we sincerely appreciate your insightful suggestions, which have helped refine our manuscript.